# Clinical Outcomes of Digital Cholangioscopy-Guided Procedures for the Diagnosis of Biliary Strictures and Treatment of Difficult Bile Duct Stones: A Single-Center Large Cohort Study

**DOI:** 10.3390/jcm10081638

**Published:** 2021-04-12

**Authors:** Hirohito Minami, Shuntaro Mukai, Atsushi Sofuni, Takayoshi Tsuchiya, Kentaro Ishii, Reina Tanaka, Ryosuke Tonozuka, Mitsuyoshi Honjo, Kenjiro Yamamoto, Kazumasa Nagai, Yukitoshi Matsunami, Yasutsugu Asai, Takashi Kurosawa, Hiroyuki Kojima, Toshihiro Homma, Takao Itoi

**Affiliations:** Department of Gastroenterology and Hepatology, Tokyo Medical University, Tokyo 160-0023, Japan; oti.hor.ih.m@gmail.com (H.M.); maezora1031@yahoo.co.jp (S.M.); a-sofuni@amy.hi-ho.ne.jp (A.S.); tsuchiya623@mac.com (T.T.); 141ishiken@gmail.com (K.I.); onakasuicyatta@yahoo.co.jp (R.T.); tonozuka1978@gmail.com (R.T.); honjo3244@yahoo.co.jp (M.H.); kenjirojiro5544@yahoo.co.jp (K.Y.); kazu4439@gmail.com (K.N.); yukitoshimatsunami1228@yahoo.co.jp (Y.M.); yacchan85@yahoo.co.jp (Y.A.); takasikurosawa716@yahoo.co.jp (T.K.); ggswh149@yahoo.co.jp (H.K.); ssautuman@gmail.com (T.H.)

**Keywords:** peroral cholangioscopy, cholangiocarcinoma, bile duct stone, accuracy, adverse event

## Abstract

Although Spy DS (SpyGlass DS Direct Visualization System) is considered to be useful for the diagnosis of bile duct strictures and the treatment of bile duct stones, there is limited data to date validating its efficacy. We hence retrospectively evaluated the clinical outcomes of the use of Spy DS in a large number of patients. A total of 183 patients who underwent Spy DS-guided procedures for indeterminate bile duct strictures (*n* = 93) and bile duct stones (*n* = 90) were analyzed retrospectively. All patients (93/93) with bile duct strictures successfully underwent visual observation, and 95.7% (89/93) of these patients successfully underwent direct biopsy. The sensitivity, specificity, and overall accuracy were 94.7%, 83.3%, and 90.3%, respectively, for visual impression; 80.9%, 100%, and 89.2%, respectively, for histopathological analysis of a direct biopsy; and 96.5%, 91.7%, and 94.6%, respectively, for visual impression combined with biopsy. Successful visualization of the stones was achieved in 98.9% (89/90) of the patients, and complete stone removal was achieved in 92.2% (83/90) of the patients, with an average of 3.3 procedures. The adverse events rate was 17.5% (32/183; cholangitis in 15 patients, fever the following day in 25, pancreatitis in 1, hemorrhage in 1, and gastrointestinal perforation in 1). No administration of antibiotics before the procedure was found to be a statistically significant risk factor for the development of fever after the procedure (*p* < 0.01). Spy DS-guided procedures are effective for the diagnosis and treatment of bile duct lesions and can be performed with a low risk of serious adverse events.

## 1. Introduction

Endoscopic retrograde cholangiopancreatography (ERCP) has been used as the primary method for the diagnosis of biliary strictures and the treatment of bile duct stones. However, the accurate visual diagnosis of biliary strictures and assessment of the extent of a lesion are not possible by ERCP, and treatment of bile duct stones by ERCP procedures is sometimes difficult. Peroral cholangioscopy (POCS) has recently become a useful modality for diagnostic and therapeutic procedures in patients with biliary diseases [1,2]. POCS enables direct visualization of the biliary tract and the sampling of suspicious lesions, together with lithotripsy with electrohydraulic lithotripsy (EHL) or laser lithotripsy. POCS may also be applied to percutaneous procedures [3,4].

Since the report by Chen and Pleskow in 2007 [5] regarding a type of POCS called the SpyGlass Legacy System (Spy Legacy, Boston Scientific Corporation, Natick, MA, USA), the system has become widely used because of its ease of use for diagnosis and treatment. Although the image quality of Spy Legacy is not as high as that of video mother-baby type POCS, it offers many features that overcome the problems of other types of POCS. In particular, the relatively high degree of freedom of operation owing to the four-directional angle, and the ability to pump and aspirate water at the same time owing to the independent irrigation ports, are very useful for accurate bile duct biopsies and enables minimally stressful lithotripsy procedures [6].

Furthermore, the new SpyGlass DS Direct Visualization System (Spy DS) was developed with marked improvements in terms of image quality and system setup compared with Spy Legacy [7]. International multicenter studies showing the efficacy and safety of the SpyGlass system for the diagnosis of biliary strictures and the treatment of difficult bile duct stones have been reported [8,9]. However, these studies included both Spy Legacy and Spy DS. Considering the differences in the professionals using Spy DS and the diversity of its uses, more data of the clinical outcomes of only Spy DS are needed to establish its usefulness in clinical practice.

Therefore, in this study, we analyzed the efficacy of Spy DS for the diagnosis and treatment of biliary diseases in a large number of patients.

## 2. Methods and Patients

### 2.1. Patients

This study included 183 consecutive patients (120 men and 63 women; median age: 69.8 years; range: 35–93 years) who underwent POCS-guided procedures using Spy DS for indeterminate bile duct strictures (*n* = 93) that could not be diagnosed by ERCP alone or for bile duct stones (*n* = 90) that were difficult to treat by ERCP alone between July 2015 and April 2020 at Tokyo Medical University Hospital. An indeterminate biliary stricture is a stricture of the bile duct that is difficult to conclude as being either benign or malignant on the basis of ERCP, bile duct biopsy under fluoroscopy, and brush cytology. A difficult common bile duct stone is a stone that could not be removed with a stone removal device, such as a basket, balloon, or mechanical lithotripter in patients who underwent a papillary procedure of endoscopic sphincterotomy (EST), endoscopic papillary balloon dilation, or endoscopic papillary large balloon dilation (EPLBD). Baseline characteristics of the patients are presented in Table 1 for those with indeterminate biliary strictures and in Table 2 for those with difficult bile duct stones. The final clinical diagnoses for which Spy DS was used for the diagnosis of bile duct strictures were as follows: hilar region cholangiocarcinoma in 33.3% (*n* = 31) of the patients, bile duct cancer in 22.6% (*n* = 21), and benign bile duct stricture in 37.6% (*n* = 35). The primary diseases for which Spy DS was used for therapeutic purposes of difficult stones were as follows: common bile duct stones in 70% (*n* = 63), intrahepatic bile duct stones in 27.8% (*n* = 25), and cholecystic stones in 2.2% (*n* = 2). The mean maximum diameter of the targeted stones was 13.6 mm. In all patients, POCS-guided procedures using Spy DS were performed after the improvement of their cholangitis. Preprocedural and postprocedural data were collected after a detailed review of the patients’ electronic medical records. Written informed consent was obtained from all patients. This retrospective study was approved by the Institutional Review Board of Tokyo Medical University (study approval no.: T2020-0349).

### 2.2. Endoscopic Procedures

#### 2.2.1. Diagnostic Procedures

The Spy DS is mainly used to differentiate between benign and malignant bile ductal strictures, which are difficult to diagnose definitively using other modalities, and to evaluate the horizontal extent of bile ductal carcinoma prior to surgical resection. The insertion of Spy DS into lesions that are difficult to pass through under fluoroscopic guidance owing to their rigid stricture, into the peripheral bile duct, or into highly bent bile ducts can be performed smoothly under guidewire guidance for the evaluation of the horizontal extent of the lesion.

After the biliary stricture is visualized, the benign or malignant nature of the lesion is determined from visual findings. Then, Spy DS-guided biopsy under direct visualization is performed using dedicated biopsy forceps, namely, Spy Bite forceps (Boston Scientific Corporation; Merlborough, MA, USA), which are small biopsy forceps with a diameter of 1 mm and a cup size of 4.2 mm. The obtained tissue specimens are immediately fixed in 10% neutral-buffered formalin solution for histological analysis by attaching the specimen directly from the Spy Bite forceps to a piece of filter paper.

#### 2.2.2. Stone Removal Procedures

After the stones were visualized under Spy DS, they were crushed with EHL and a holmium YAG laser. EHL works on the principle that sparks discharged under water generate high-frequency hydraulic pressure waves. The high-pulse energy used to disintegrate the stones may damage the bile duct wall, and therefore, cholangioscopic visualization of the target stones is mandatory in EHL therapy. The YAG laser has a straight-line shock wave propagation, resulting in crushing at a single point while minimally affecting the bile duct wall. This laser is easy to use as the probe tip is pressed against the stone. However, it takes a considerable amount of time to crush the stone because the probe can only be applied directly to the area where the probe is pressed against the stone. In addition, the laser itself is very expensive, and therefore the number of facilities where it can be used is limited. In our hospital, we first use EHL for lithotripsy, and then for stones that are too hard to break, we use the YAG laser that has superior straight-line crushing power to EHL. The type of EHL and YAG laser are appropriately selected according to the characteristics of the stones.

### 2.3. Outcome Measurements

In this study, several aspects of POCS-guided procedures using Spy DS, divided mainly into the diagnostic procedures for bile duct strictures and stone removal procedures for difficult stones were analyzed. Methods of bile duct intubation, final drainage, administration of antibiotics before and after the procedure, vital signs during and after the procedure, blood test data, and adverse events were commonly assessed. Regarding diagnostic procedures, technical success was defined as the visualization of a bile duct stricture. The visual diagnosis of a benign stricture was based on the endoscopic appearance of a smooth surface and outline with no visible abnormal vessels (Figure 1a). The visual diagnosis of a malignant stricture was based on the endoscopic appearance of villous mucosal projections, irregular mucosal nodularity, mass-forming lesions, and prominent vascularization (Figure 1b). The final clinical diagnosis of a benign stricture was based on a negative result for malignancy upon biopsy analysis, and clinical data indicating no deterioration at the 6-month follow-up. The final clinical diagnosis of a malignant stricture was based on confirmation of malignancy upon histological analysis of the biopsies or surgically resected specimens, or disease progression at the 6-month follow-up. The processing and evaluation of the tissue specimens obtained by biopsy of the bile duct were according to our institution’s standards. Briefly, tissue sections were stained with hematoxylin and eosin for evaluation by a pathologist. Immunohistochemical procedures were performed if necessary. Accurate diagnosis using visual findings on Spy DS or histological findings obtained by biopsy was defined as an accurate differentiation between malignant and benign biliary strictures.

Regarding stone removal procedures using Spy DS-guided lithotripsy, the technical success of a therapeutic procedure was defined as the visualization of bile duct stones and the crushing of stones using EHL or the YAG laser. The clinical success of a therapeutic procedure was defined as complete removal of bile duct stones. The location and diameter of the stones, papillary procedures prior to Spy DS insertion, additional mechanical lithotripsy methods, and the number of ERCPs required for complete stone removal were analyzed.

Regarding procedure-associated adverse events, they were graded according to the severity grading system of the American Society for Gastrointestinal Endoscopy lexicon [10]. Fever associated with the procedure was defined as a fever of higher than 38.0 °C on the day after the procedure. The diagnosis of cholangitis associated with procedures using Spy DS was in accordance with the Tokyo Guidelines 2018 [11].

### 2.4. Statistical Analysis

Continuous variables were presented as the mean ± standard deviation or the median with range and were compared using the Student *t*-test or the Wilcoxon rank sum test as appropriate. Categorical variables were compared using the Chi-squared or Fisher exact test. Statistical analyses were performed using Statistical Package for the Social Sciences software version 26 (IBM; Chicago, IL, USA). A *p*-value of less than 0.05 was considered to indicate a statistically significant difference between groups.

## 3. Results

### 3.1. Diagnostic Procedures

Clinical outcomes of diagnostic Spy DS are shown in Table 3. Spy DS was inserted into the bile duct with a guidewire in 94.6% (*n* = 88) of patients and was inserted wire-free in 5.4% (*n* = 5) of patients. After the examination, draining was performed by endoscopic nasobiliary drainage (ENBD) in 43.0% (*n* = 40) of the patients, by endoscopic biliary drainage (EBD) in 41.9% (*n* = 39), and 15.1% (*n* = 14) of the patients did not require drainage. Regarding adverse events, fever, cholangitis, and EST hemorrhage occurred in 10.8% (*n* = 10), 4.3% (*n* = 4), and 1.1% (*n* = 1) of the patients, respectively. Visualization of the stricture was successful in all patients (93/93). The diagnostic accuracies of visual impressions by Spy DS and of histopathological analysis of direct biopsies are shown in Table 4. The diagnostic yields of visual impression had a sensitivity of 94.7%, specificity of 83.3%, a positive predictive value (PPV) of 90.0%, and a negative predictive value (NPV) of 90.9%, with an overall accuracy of 90.3%. On the other hand, the diagnostic yields of histopathological analysis of direct biopsies had a sensitivity of 80.9%, specificity of 100%, PPV of 100%, NPV of 80.0%, and an overall accuracy of 89.2%. Surgical resection was performed in 48.4% (45/93) of the patients after assessment of disease extension by direct visual mapping biopsy using Spy DS.

### 3.2. Stone Removal Procedures

The clinical outcomes of therapeutic Spy DS are shown in Table 5. Technical success was achieved in 98.9% (89/90) of the patients. In one patient, obtaining a frontal view of the bile duct stones was difficult. Of the patients in whom stone removal was attempted using POCS-guided EHL or the YAG laser with Spy DS, difficulty in stone removal was experienced in 7.8% (*n* = 7) of the patients because of difficulty in viewing the stones frontally or because the stones were in the peripheral bile duct and were hence difficult to approach. There were two patients with severe bile duct stenosis in which the Spy DS could not be passed through even after balloon dilation, four patients in whom intrahepatic resection was performed owing to the inability to approach the peripheral intrahepatic stones, and one patient who required periodic stent replacement owing to difficulties in lithotripsy despite the use of multiple EHLs. Ultimately, the clinical success rate of the therapeutic procedure was 92.2%. We cannulated Spy DS into the bile duct with a guidewire in 93.3% (*n* = 84) of the patients and by the wire-free procedure in 6.7% (*n* = 6) of the patients. Regarding papillary treatment for stone removal, only EST was used in 65.6% (*n* = 59) of the patients, and EPLBD was performed in 34.4% (*n* = 31) of the patients. Under Spy DS guidance, EHL was used for lithotripsy in 97.8% (*n* = 88) of the patients, and a YAG laser was used in 2.2% (*n* = 2) of the patients. After lithotripsy using EHL or the YAG laser, the devices used for stone removal were baskets in 70% (*n* = 63), balloons in 53.3% (*n* = 48), and mechanical lithotripters in 58.9% (*n* = 53) of the patients. The method of drainage after the procedure was ENBD in 52.2% (*n* = 47), EBD in 25.6% (*n* = 23), and was performed drainage-free in 21.1% (*n* = 19) of the patients. An average of 3.3 ERCPs were required for complete stone removal, including stone removal using EHL or the YAG laser. Regarding adverse events, fever occurred in 16.7% (*n* = 15), cholangitis in 12.2% (*n* = 11), pancreatitis in 1.1% (*n* = 1), and gastrointestinal perforation in 1.1% (*n* = 1) of the patients.

### 3.3. Risk Factors of Cholangitis and Fever after Spy DS-Guided Procedures

We also analyzed the risk factors of cholangitis and fever after the procedures. The risk factors for cholangitis are shown in Table 6. Although no statistically significant risk factor was identified, cholangitis after the procedure tended to occur in patients with no administration of antibiotics before the procedure and those with benign biliary strictures. On the other hand, risk factors for fever on the day after the procedure are shown in Table 7. No administration of antibiotics before the procedure was found to be a statistically significant risk factor of fever after the procedure (*p* < 0.01).

## 4. Discussion

To the best of our knowledge, our present study is the first to determine the efficacy and safety of the use of Spy DS alone for the diagnosis of indeterminate biliary strictures and the treatment of difficult bile duct stones by analyzing a large number of patients. The clinical outcomes of the patients showed that the use of Spy DS resulted in a high diagnostic accuracy rate and a high treatment success rate, with a low rate of adverse events.

Despite improvements in noninvasive diagnostic modalities, such as computed tomography (CT) and magnetic resonance imaging (MRI), the diagnosis of biliary strictures is often difficult. Histological evidence is needed to confirm the diagnosis, so ERCP is performed, and forceps biopsy and brushing cytology are performed under fluoroscopy. The accuracy of biliary tract biopsy under ERCP is not very high and has been reported to range from 41% to 92% [12,13]. Therefore, diagnosis needs to be performed in conjunction with EUS and intraductal ultrasonography (IDUS) results. Furthermore, it is difficult to assess tumor extension with ERCP, although this is very important for determining the indication for surgical resection of bile duct cancers in the hilar region. On the other hand, the POCS-guided procedure enables biopsies to be performed under direct visualization of the bile duct mucosa and hence enables a more targeted biopsy as well as evaluation on the basis of POCS findings. Among POCS, SpyGlass Legacy (fiberglass POCS) has been reported to have a visual diagnostic ability of 80% to 94% [6,14]. This device is easy to operate, but its weakness is its low image resolution. Image resolution was improved by the development of Spy DS (an electronic scope), leading to the improvement of diagnostic ability. Among previous reports, Navaneethan et al. reported a retrospective multicenter clinical study of the use of Spy DS for the diagnosis of pancreaticobiliary disorders. In their study, diagnostic Spy DS was carried out in 44 patients. The sensitivity and specificity for the detection of malignancy based on visual findings were 90% (95% confidence interval (CI): 69.9–97.2) and 95.8% (95% CI: 79.8–99.3), respectively [15]. In another study, Alan et al. reported the overall accuracy of visual analysis using Spy DS for differentiating malignant from benign ductal strictures to be 97.2% [16]. In the present study, the sensitivity, specificity, and accuracy of the visual findings using Spy DS were 94.7%, 83.3%, and 90.8%, respectively. All of the studies have different backgrounds, and therefore they cannot be accurately compared, but the visual diagnostic performance of Spy DS was found to be higher than that of SpyGlass Legacy. Regarding the diagnostic ability of bile duct biopsies performed under Spy DS, the sensitivity and specificity for the diagnosis of malignancy based on forceps biopsy under Spy DS were reported to be 85% (95% CI: 64.0–94.8) and 100% (95% CI: 86.2–100), respectively [15]. In other studies, the diagnostic accuracy using Spy Bite forceps under Spy DS was 72.2% [16]. In the present study, the sensitivity and specificity of diagnosis by direct biopsy were 80.9% and 100%, respectively. Although the specificity of direct biopsy is higher than that of visual analysis, its sensitivity is lower. Table 8 shows a summary of the studies on the characteristics and outcomes of all diagnostic procedures performed by POCS. These data suggest that complementary diagnosis by bile duct biopsy is required in addition to visual analysis. Although visual diagnosis using Spy DS has been found to have a high diagnostic accuracy in patients with indeterminate biliary strictures, at present, there are no clear criteria or objective assessment systems for the visual diagnosis of a lesion. “Endoscopic impression” under visualization is actually a clinical impression of the physician who is informed of the patient’s history, as well as the patient’s physical, radiographical, and laboratory results. If a visual diagnosis of malignancy is suspected by Spy DS, and furthermore the patient’s history, physical examination, blood tests, and other imaging analyses suggest malignancy, the lesion may be surgically resected after discussion with the patient and the multidisciplinary team, including radiologists, oncologists, surgeons, and gastroenterologists. In the present study, of the patients in whom malignancy was suspected on visual findings using Spy DS and other tests, but in whom no malignancy was detected on histological analysis of biopsy specimens, malignancy was detected in surgical resection specimens with a probability of 88.9% (8/9). The accuracy of Spy Bite biopsies was lower than that of visual diagnosis, which could be attributed to the difficulty in obtaining tissue samples owing to the tangential location of the stricture to the Spy DS, and the inadequate volume of the samples, even when they could be obtained. In this study, 10.1% (31/306) of the biopsies taken with Spy Bite had an insufficient sample volume. Therefore, further studies to establish objective macroscopic grading criteria of malignant features and the improvement of biopsy forceps are warranted.

Regarding the therapeutic ability of Spy DS for bile duct stones, some studies have reported stone removal using POCS-guided EHL with an 86% to 100% stone removal rate [15,17,18]. Table 9 shows a summary of the studies analyzing the characteristics and outcomes of stone removal procedures by POCS. We achieved complete stone removal in 92.2% (83/90) of the patients using only Spy DS. In general, intrahepatic bile duct stones are more difficult to treat than large common bile duct stones. Our study included 25 patients with intrahepatic bile duct stones, and the success rate of treatment was 84% (21/25). We consider that the improved operability and functionality of Spy DS (four channels, high image resolution, and highly efficient water pumping and aspiration functions) contributed to the favorable treatment outcomes. However, in some patients in whom the peripheral intrahepatic bile duct was filled with stones, surgical intervention was ultimately necessary because the stone removal instruments could not reach the stones even using Spy DS.

We believe that diagnosis and treatment using POCS is an area that will develop further in the future. Spy DS has excellent diagnostic ability based on visual impressions. To further improve its diagnostic ability, the image quality needs to be improved by using imaging techniques, such as narrow-band imaging (NBI) and magnification. On the other hand, to improve the diagnostic ability based on the histopathological analysis of direct biopsies, it is necessary to develop biopsy forceps that can collect a sufficient amount of specimen.

At present, other types of POCS methods are available, including video mother-baby type POCS and direct POCS. Video mother-baby type POCS uses a duodenoscope channel to introduce a small cholangioscope into the bile duct [21]. This type of POCS has high image quality but has some limitations. For example, it is impossible to perform operations, such as washing, aspiration, and biopsy, at the same time because the device has only one channel. Furthermore, the degree of freedom of manipulation in the bile duct is low because only two directions of manipulation are possible. The device is also very fragile and requires frequent repairs. On the other hand, direct POCS provides high image quality and enables detailed evaluation [22]. The large diameter of the working channel makes it easy to insert various instruments, such as biopsy forceps, into the working channel. Furthermore, compared to video mother-baby type POCS, direct POCS has the advantage that it can be performed by a single endoscopist, and the device is less fragile. However, the most crucial problem of direct POCS is the difficulty of insertion of the device into the bile duct. Since each has its own advantages and disadvantages, the issue to be addressed in the future is how to use SpyDS and the other POCS methods properly.

It has been reported that the most common procedure-associated adverse event of POCS is cholangitis, which occurred in 0% to 14% of procedures [23]. We analyzed risk factors for cholangitis, and although there was no statistically significant association between prophylactic antibiotics administration and the occurrence of cholangitis after a procedure, the results show that antibiotics may be useful for the prevention of cholangitis. The high occurrence of cholangitis (10.6%) reported in another study without prophylactic antimicrobials administration also supports our results [24]. In the present study, apart from cholangitis, the procedure-associated adverse event of fever without liver dysfunction was additionally assessed. This is predicted to occur due to transient bacteremia, in which bacteria are refluxed into the bloodstream owing to the increased intrabiliary pressure associated with irrigation during the Spy DS procedure. The results of our study suggest that prophylactic antibiotics administration is useful for the prevention of fever after the procedure. We suggest that this is because prophylactic antibiotics administration prevents or treats transient bacteremia.

As for other adverse events, post-ERCP pancreatitis is a commonly known adverse event of ERCP. According to a systematic survey, it is reported that the cumulative incidence of post ERCP pancreatitis was 3.47% [25]. However, there was only one patient (0.5%) who developed moderate pancreatitis as an adverse event of POCS procedure in this study. This is because in many of the cases included in this study ERCP and EST had been already performed prior to POCS procedures. Therefore, the risk of post-ERCP pancreatitis was low in the cases of this study.

Our study has several limitations. Because this study is a retrospective and single-center study, technical and selection biases are inevitable. The criteria for the visual impressions of endoscopists have not been established, and there were no clear criteria for the administration of antibiotics before the procedure. To date, only a few case reports and case series of diagnostic and therapeutic procedures using Spy DS have been reported [26,27,28,29,30,31]. To increase the visual diagnostic performance of Spy DS, validation of the objective classification of the virtual criteria is needed. Therefore, further prospective multicenter studies should be performed in the future.

In conclusion, Spy DS-guided procedures are effective for the diagnosis and treatment of bile duct lesions and can be performed with a low risk of serious adverse events. Furthermore, prophylactic antibiotics administration may be useful for the prevention of cholangitis and fever after a procedure.

## Figures and Tables

**Figure 1 jcm-10-01638-f001:**
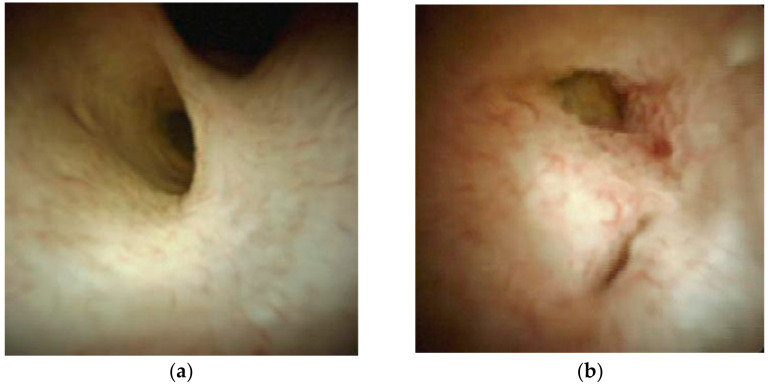
Indeterminate biliary strictures identified using Spy DS (SpyGlass DS Direct Visualization System). (**a**) A benign stricture showing a smooth surface and outline with no visible abnormal vessels. (**b**) A malignant stricture showing villous mucosal projections, irregular mucosal nodularity, and prominent vascularization.

**Table 1 jcm-10-01638-t001:** Characteristics of patients with indeterminate biliary strictures.

Variable		Patients (*n* = 93)
Age, median (range), years		69 (38–87)
Sex, female/male, *n*		25/68
Final diagnosis, *n* (%)	Hilar region cholangiocarcinoma	31 (33.3)
	Bismuth I/II/IIIa/IIIb/IV	4/3/7/4/13
	Bile duct cancer	21 (22.6)
	upper/middle/lower	3/7/11
	Cholecystic duct cancer	4 (4.3)
	IgG4-SC	4 (4.3)
	PSC	5 (5.4)
	IPNB	2 (2.2)
	Benign bile duct stricture	26 (28.0)

IgG4-SC, IgG4-related sclerosing cholangitis; PSC, primary sclerosing cholangitis; IPNB, intraductal papillary neoplasm of the bile duct.

**Table 2 jcm-10-01638-t002:** Characteristics of patients with difficult bile duct stones.

Variable		Patients (*n* = 90)
Age, median (range), years		72 (35–93)
Sex, female/male, *n*		38/52
Final diagnosis, *n* (%)	Intrahepatic bile duct stones	25 (27.8)
	Common bile duct stones	63 (70)
	Cystic duct stones	2 (2.2)
Mean maximum diameter of the targeted stones ± SD (mm)		13.6 ± 7.01

SD, standard deviation.

**Table 3 jcm-10-01638-t003:** Clinical outcomes of diagnostic procedures by Spy DS.

Variable		Patients (*n* = 93)
Diagnostic devices in combination with Spy DS, *n* (%)	IDUS	19 (20.4)
	EUS	3 (3.2)
Bile duct cannulation method, *n* (%)	Using a guidewire	88 (94.6)
	Guidewire-free	5 (5.4)
Method of drainage after the procedure, *n* (%)	ENBD	40 (43.0)
	EBD	39 (41.9)
	Drainage-free	14 (15.1)
Success rate of stricture visualization, *n* (%)		93 (100)
Adverse events, *n* (%)		
Mild	Fever	10 (10.8)
Moderate	Cholangitis	4 (4.3)
	Post-EST hemorrhage	1 (1.1)

IDUS, intraductal ultrasonography; EUS, endoscopic ultrasonography; ENBD, endoscopic nasobiliary drainage; EBD, endoscopic biliary drainage; EST, endoscopic sphincterotomy.

**Table 4 jcm-10-01638-t004:** Diagnostic yields of visualization by Spy DS and histopathological analysis of direct biopsies.

	Diagnosis Based on Visual Impression (*n* = 93)	Diagnosis Based on Histopathological Analysis of a Direct Biopsy (*n* = 93)	Diagnosis Based on Visual Impression Combined with Biopsy (*n* = 93)
Sensitivity (%)	94.7	80.9	96.5
Specificity (%)	83.3	100	91.7
Positive predictive value (%)	90.0	100	94.8
Negative predictive value (%)	90.9	80	94.3
Overall accuracy (%)	90.3	89.2	94.6

**Table 5 jcm-10-01638-t005:** Clinical outcomes of therapeutic Spy DS.

Variable		Patients (*n* = 90)
Technical success, *n* (%)		89 (98.9)
Clinical success, *n* (%)		83 (92.2)
Bile duct cannulation method, *n* (%)	Using a guidewire	84 (93.3)
	Guidewire-free	6 (6.7)
Papillary treatment before the cannulation, *n* (%)	EST	59 (65.6)
	EPLBD with EST	31 (34.4)
Devices used for stone removal after lithotripsy, *n* (%)	Basket	63 (70)
	Balloon	48 (53.3)
	Mechanical lithotripter	53 (58.9)
Method of drainage after the procedure, *n* (%)	ENBD	47 (52.2)
	EBD	23 (25.6)
	Drainage-free	19 (21.1)
Number of ERCPs required for complete stone removal, mean ± SD		3.3 ± 2.38
Adverse events, *n* (%)		
Mild	Fever	15 (16.7)
Moderate	Cholangitis	11 (12.2)
	Pancreatitis	1 (1.1)
Severe	Intestinal perforation	1 (1.1)

EPLBD, endoscopic papillary large balloon dilation.

**Table 6 jcm-10-01638-t006:** Analysis of risk factors for cholangitis.

	Cholangitis (+) (*n* = 15)	Cholangitis (−) (*n* = 168)	*p*-Value
Antibiotics administration (+) (*n* = 87)	4	83	*p* = 0.09
Antibiotics administration (−) (*n* = 96)	11	85	
Malignant disease (*n* = 56)	1	55	*p* = 0.07
Benign disease (*n* = 127)	14	113	
Diagnostic procedures (*n* = 93)	4	89	*p* = 0.09
Stone removal procedures (*n* = 90)	11	79	
Drainage with ENBD or EBD (*n* = 150)	12	138	*p* = 0.89
Drainage-free (*n* = 33)	3	30	

ENBD, endoscopic nasobiliary drainage; EBD, endoscopic biliary drainage; +, complications of cholangitis; −, no complications of cholangitis.

**Table 7 jcm-10-01638-t007:** Analysis of risk factors for a fever of higher than 38.0 °C on the day after the procedure.

	Fever (+) (*n* = 25)	Fever (−) (*n* = 158)	*p*-Value
Antibiotics administration (+) (*n* = 87)	4	83	*p* < 0.01
Antibiotics administration (−) (*n* = 96)	21	75	
Malignant disease (*n* = 56)	6	50	*p* = 0.44
Benign disease (*n* = 127)	19	108	
Diagnostic procedures (*n* = 93)	10	83	*p* = 0.24
Stone removal procedures (*n* = 90)	15	75	
Drainage with ENBD or EBD (*n* = 150)	22	128	*p* = 0.4
Drainage-free (*n* = 33)	3	30	

ENBD, endoscopic nasobiliary drainage; EBD, endoscopic biliary drainage.

**Table 8 jcm-10-01638-t008:** Summary of studies on the characteristics and outcomes of diagnostic procedures by POCS.

Author	Year	Study Design	Cholangioscope	*n*	Diagnostic Method	SE, %	SP, %	PPV, %	NPV, %	AC, %	AE, %
Chen et al [5].	2011	Multicenter retrospective	Spy Legacy	95	Visual	78	82	80	80	80	NA
				95	Biopsy	49	98	100	72	75	NA
Kurihara et al [6].	2016	Multicenter prospective	Spy Legacy	84	Visual	95	93	96	89	94	NA
				73	Biopsy	65	100	73	100	73	NA
Navaneethan et al [15].	2016	Multicenter retrospective	Spy DS	44	Visual	90	96	95	92	93	2.9
				44	Biopsy	85	100	100	89	93	2.9
Ogura et al [17].	2017	Single-center retrospective	Spy DS	30	Visual	83	89	83	89	87	6
				28	Biopsy	80	100	100	90	93	6
Almadi et al [8].	2020	Multicenter retrospective	Spy Legacy Spy DS	290	Visual	86.7	71.2	65.8	89.4	77.2	NA
				163	Biopsy	75.3	100	100	77.1	86.5	NA
Present study	2021	Single-center retrospective	Spy DS	93	Visual	94.7	83.3	90.0	90.9	90.3	5.4
				93	Biopsy	80.9	100	100	80	89.2	5.4

POCS, peroral cholangioscopy; SE, sensitivity; SP, specificity; PPV, positive predictive value; NPV, negative predictive value; AC, accuracy; AE, adverse event; NA, not available.

**Table 9 jcm-10-01638-t009:** Summary of studies on the characteristics and outcomes of stone removal procedures by POCS.

Author	Year	Study Design	Cholangioscope	*n*	Lithotripsy	Clearance, %	Mean No. of Procedures	Mean Size of Stone, mm	AE, %
Kurihara et al [6].	2016	Multicenter prospective	Spy Legacy	31	EHL, LL	74.2	1.9	20.6	NA
Navaneethan et al [15].	2016	Multicenter retrospective	Spy DS	31	LL	97.2	1.1	14.9	NA
Turowski et al [19].	2018	Multicenter retrospective	Spy DS	107	EHL, LL	91.1	3	NA	NA
Maydeo et al [9].	2019	Multicenter retrospective	Spy Legacy Spy DS	156	EHL, LL	87.0	NA	18.0	2.5
Bokemeyer et al [20].	2020	Multicenter retrospective	Spy DS	60	EHL, LL	95.0	NA	20.0	NA
Present study	2021	Single-center retrospective	Spy DS	90	EHL, LL	92.2	3.3	13.6	14.4

EHL, electrohydraulic lithotripsy; LL, laser lithotripsy.

## Data Availability

Authors should accurately present their research findings and include an objective discussion of the significance of their findings.

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
