# Peer review of "Clinical Outcomes of Digital Cholangioscopy-Guided Procedures for the Diagnosis of Biliary Strictures and Treatment of Difficult Bile Duct Stones: A Single-Center Large Cohort Study"

_jcm, 2021, doi:10.3390/jcm10081638_

Round 1
Reviewer 1 Report
In this study, authors retrospectively evaluated the clinical outcomes of the use of Spy DS in a large number of patients. Although this was the retrospective analysis, the study results were well organized and statistically analyzed.
Major
- The definition of indeterminate biliary stricture and difficult CBD stone has not been clearly presented. Please add the definition of indeterminate biliary stricture and difficult CBD stone in the “Methods”.
- In risk factors for development of fever after POC using SpyGlass DS, multivariate analysis was not performed. Please review whether there is any statistical problem with establishing antibiotics administration as a risk factor for fever after POC.
- And, did you performed POC using SpyGlass DS after improving prior cholangitis in all patients? In order to further clarify the risk factor for development of cholangitis and/or fever after POC, it would be better to add that the POC was performed after the improvement of cholangitis.
Minor
- Please indicate the “standard deviation” at line of “Mean maximum diameter of the targeted stones” in Table 2.
- In table 5: Number of ERCPs required for complete stone removal (n) : mean or median?
Author Response
Point-by-point responses to the comments by Reviewer 1
First, we would like to thank Reviewer 1 for the valuable comments regarding our manuscript. Our responses to each of the comments are as follows.
Major comment
- The definition of indeterminate biliary stricture and difficult CBD stone has not been clearly presented. Please add the definition of indeterminate biliary stricture and difficult CBD stone in the “Methods”.
Response:
In accordance with the comment, we added the definition of intermediate biliary stricture and difficult CBD stone to the Methods and patients section of the revised manuscript, as follows. (page 7, lines 7–14)
“An indeterminate biliary stricture is a stricture of the bile duct that is difficult to conclude as being either benign or malignant on the basis of ERCP, bile duct biopsy under fluoroscopy, and brush cytology. A difficult common bile duct stone is a stone that could not be removed with a stone removal device, such as a basket, balloon, or mechanical lithotripter in patients who underwent a papillary procedure of endoscopic sphincterotomy (EST), endoscopic papillary balloon dilation, or endoscopic papillary large balloon dilation (EPLBD).”
- In risk factors for development of fever after POC using SpyGlass DS, multivariate analysis was not performed. Please review whether there is any statistical problem with establishing antibiotics administration as a risk factor for fever after POC.
Response:
We thank you very much for your valuable advice.
We did not perform multivariate analysis of fever after POCS using SpyGlass DS, because no other factor was found to be a significant risk factor in univariate analysis when we analyzed other factors. Therefore, we believe there are no statistical problems with our analysis.
- And, did you performed POC using SpyGlass DS after improving prior cholangitis in all patients? In order to further clarify the risk factor for development of cholangitis and/or fever after POC, it would be better to add that the POC was performed after the improvement of cholangitis.
Response:
We thank you very much for this comment.
In our hospital, we do not perform cholangioscopy in patients who have cholangitis, and hence all of the patients analyzed by cholangioscopy in this study did not have cholangitis.
To make this point clear, we added the following sentence to the Methods and patients section of the revised manuscript. (page 8, lines 4–5)
“In all patients, POCS-guided procedures using Spy DS were performed after the improvement of their cholangitis.”
Minor comment
- Please indicate the “standard deviation” at line of “Mean maximum diameter of the targeted stones” in Table 2.
Response:
In accordance with the comment, we added the standard deviation to the relevant section of Table 2 in the revised manuscript.
- In table 5: Number of ERCPs required for complete stone removal (n) : mean or median?
Response:
Thank you for pointing out our mistake. The figures and tables show the mean values, and we have indicated this as appropriate in the revised manuscript.
Reviewer 2 Report
I would like to commend the authors on this research.
Before considering the publication, I put some comments and questions:
- Introduction is too long. The paper describes the results on the use of SpyGlass DS. I consider the review about the different types of peroral cholangioscopy systems not useful in this context.
- The authors should include a definition for difficult bile duct stones;
-
How was EHL and YAG laser selected. The authors should explain the rationale for choosing one technology over the other, or exclude the only two cases regarding the use of YAG laser;
-
Regarding adverse events post ercp pancreatitis should be reported;
-
Surgical resection was performed in 48.4% (45/93) of the patients after assessment of disease progression by direct visual mapping biopsy using Spy DS. Do the authors mean extension?
- YAG laser was used only in two cases of stone disease vs 88 EHL cases. For the purpuse of data homogeneity these two cases should be excluded.
-
Please consider excluding the two paragraphs on the discussion from line 329 to 340.
-
In the summary of studies on diagnostic POCS the authors missed one study that is worth mentioning Dig. Dis 2020;38(5):431-440 doi: 10.1159/000504910.
Author Response
Point-by-point responses to the comments by Reviewer 2
First, we would like to thank Reviewer 2 for the valuable comments regarding our manuscript. Our responses to each of the comments are as follows.
Introduction is too long. The paper describes the results on the use of SpyGlass DS. I consider the review about the different types of peroral cholangioscopy systems not useful in this context.
Response:
We agree that our Introduction section was too long. Therefore, we have deleted the description of the characteristics of several types of POCS methods, including video mother-baby type POCS and direct POCS from the manuscript, as the main focus of this study is the results of the use of SpyGlass DS.
The authors should include a definition for difficult bile duct stones.
Response:
In accordance with the comment, we have added the definitions of indeterminate biliary stricture and difficult bile duct stone to the Methods and patients section of the revised manuscript, as follows. (page 7, lines 7–14)
“An indeterminate biliary stricture is a stricture of the bile duct that is difficult to conclude as being either benign or malignant on the basis of ERCP, bile duct biopsy under fluoroscopy, and brush cytology. A difficult common bile duct stone is a stone that could not be removed with a stone removal device, such as a basket, balloon, or mechanical lithotripter in patients who underwent a papillary procedure of endoscopic sphincterotomy (EST), endoscopic papillary balloon dilation, or endoscopic papillary large balloon dilation (EPLBD).”
How was EHL and YAG laser selected. The authors should explain the rationale for choosing one technology over the other, or exclude the only two cases regarding the use of YAG laser.
Response:
Basically, EHL is our first choice for lithotripsy, and for hard stones that cannot be broken by EHL, the YAG laser is used because it has superior straight-line crushing power to EHL.
We have added this information to the stone removal procedures section of the revised manuscript, as follows. (page 10, lines 4–6)
“In our hospital, we first use EHL for lithotripsy, and then for stones that are too hard to break, we use the YAG laser that have superior straight-line crushing power to EHL.”
Regarding adverse events post ERCP pancreatitis should be reported.
Response:
Thank you for this important comment. We added information regarding adverse events to the Discussion section of the revised manuscript, as follows. (page 22, lines 3–9)
“As for other adverse events post-ERCP pancreatitis is a commonly known adverse event of ERCP. According to a systematic survey it is reported that the cumulative incidence of post ERCP pancreatitis was 3.47%.
(Andriulli A, Loperfido S, Napolitano G et al. Incidence rates of post-ERCP complications: a systematic survey of prospective studies. Am J Gastroenterol 2007; 102: 1781-1788)
However, there was only one patient (0.5%) who developed moderate pancreatitis as an adverse event of POCS procedure in this study. This is because in many of the cases included in this study ERCP and EST had been already performed prior POCS procedures. Therefore, the risk of post-ERCP pancreatitis was low in the cases of this study.”
Surgical resection was performed in 48.4% (45/93) of the patients after assessment of disease progression by direct visual mapping biopsy using Spy DS. Do the authors mean extension?
Response:
We apologize for our mistake. We assessed the extension of the lesion with POCS. We have corrected “progression” to “extension” in the revised manuscript.
YAG laser was used only in two cases of stone disease vs 88 EHL cases. For the purpuse of data homogeneity these two cases should be excluded.
Response:
We thank you for your comment. Considering the purpose of date homogeneity, as you mention, the case of the YAG laser may be excluded. However, the YAG laser was only used in two patients, and so we think this has little effect on the data homogeneity. We would like to continue to include these two cases.
Please consider excluding the two paragraphs on the discussion from line 329 to 340.
Response:
In accordance with the comment, we deleted these paragraphs from the manuscript, as they are not relevant to our study, which is aimed at the diagnosis and treatment outcome of POCS.
In the summary of studies on diagnostic POCS the authors missed one study that is worth mentioning Dig. Dis 2020;38(5):431-440 doi: 10.1159/000504910.
Response:
We thank you very much for this suggestion. We have included this study in the Discussion section, as follows.
Reviewer 3 Report
Dear Editor, thank you so much for inviting me to revise this manuscript.
The study addresses a current topic.
The manuscript is quite well written and organized. English could be improved.
Figures and tables are comprehensive and clear.
The introduction explains in a clear and coherent manner the background of this study.
We suggest the following modifications:
- Introduction section: although the authors correctly included important papers in this setting, we believe a couple of studies should be cited within the introduction (doi: 10.1111/den.13935; doi: 10.21873/invivo.11964.), only for a matter of consistency. We think it might be useful to introduce the topic of this interesting study.
- Methods and Statistical Analysis: nothing to add.
- Discussion section: Very interesting and timely discussion. Of note, the authors should expand the Discussion section, including a more personal perspective to reflect on. For example, they could answer the following questions – in order to facilitate the understanding of this complex topic to readers: what potential does this study hold? What are the knowledge gaps and how do researchers tackle them? How do you see this area unfolding in the next 5 years? We think it would be extremely interesting for the readers.
However, we think the authors should be acknowledged for their work. In fact, they correctly addressed an important topic, the methods sound good and their discussion is well balanced.
One additional little flaw: the authors could better explain the limitations of their work, in the last part of the Discussion.
We believe this article is suitable for publication in the journal although major revisions are needed. The main strengths of this paper are that it addresses an interesting and very timely question and provides a clear answer, with some limitations.
We suggest a linguistic revision and the addition of some references for a matter of consistency. Moreover, the authors should better clarify some points.
Author Response
Point-by-point responses to the comments by Reviewer 3
First, we would like to thank Reviewer 3 for the valuable comments regarding our manuscript. Our responses to each of the comments are as follows.
Introduction section: although the authors correctly included important papers in this setting, we believe a couple of studies should be cited within the introduction (doi: 10.1111/den.13935; doi: 10.21873/invivo.11964.), only for a matter of consistency. We think it might be useful to introduce the topic of this interesting study.
Response:
We thank you for this suggestion. We have included these studies to the Introduction section of the revised manuscript, as follows.
Discussion section: Very interesting and timely discussion. Of note, the authors should expand the Discussion section, including a more personal perspective to reflect on. For example, they could answer the following questions – in order to facilitate the understanding of this complex topic to readers: what potential does this study hold? What are the knowledge gaps and how do researchers tackle them? How do you see this area unfolding in the next 5 years? We think it would be extremely interesting for the readers.
Response:
We thank you for your valuable advice. We believe that diagnosis and treatment using POCS is an area that will develop further in the future. In accordance with the advice, we added the following description to the Discussion section of the revised manuscript. (page 20, lines 3–page 21, lines 6)
“We believe that diagnosis and treatment using POCS is an area that will develop further in the future. Spy DS, has excellent diagnostic ability based on visual impressions. To further improve its diagnostic ability, the image quality needs to be improved, by using imaging techniques, such as narrow-band imaging (NBI) and magnification. On the other hand, to improve the diagnostic ability based on the histopathological analysis of direct biopsies, it is necessary to develop biopsy forceps that can collect a sufficient amount of specimen.
At present, other types of POCS methods are available, including video mother-baby type POCS and direct POCS. Video mother-baby type POCS uses a duodenoscope channel to introduce a small cholangioscope into the bile duct [Ishida, Y.; Itoi, T.; Okabe, Y. Types of Peroral Cholangioscopy: How to Choose the Most Suitable Type of Cholangioscopy. Curr Treat Options Gastroenterol. 2016 Jun;14(2):210-9.]. This type of POCS has high image quality, but has some limitations. For example, it is impossible to perform operations, such as washing, aspiration, and biopsy at the same time because the device has only one channel. Furthermore, the degree of freedom of manipulation in the bile duct is low because only two directions of manipulation are possible. The device is also very fragile and requires frequent repairs. On the other hand, direct POCS provides high image quality and enables detailed evaluation [Itoi T, Nageshwar Reddy D, et al. Clinical evaluation of a prototype multi-bending peroral direct cholangioscope. Dig Endosc. 2014 Jan;26(1):100-7.]. The large diameter of the working channel makes it easy to insert various instruments, such as biopsy forceps, into the working channel. Furthermore, compared to video mother-baby type POCS, direct POCS has the advantage that it can be performed by a single endoscopist, and the device is less fragile. However, the most crucial problem of direct POCS is the difficulty of insertion of the device into the bile duct. Since each has its own advantages and disadvantages, the issue to be addressed in the future is how to use SpyDS and the other POCS methods properly.”
One additional little flaw: the authors could better explain the limitations of their work, in the last part of the Discussion.
Response:
In accordance with the comment, we included the limitations of our study to the Discussion section of the revised manuscript, as follows. (page 23, lines5–7)
“The criteria for the visual impressions of endoscopists have not been established, and there were no clear criteria for the administration of antibiotics before the procedure.”
Round 2
Reviewer 3 Report
The authors modified the paper, according to our suggestions.
However, they did not consider this point:
"Introduction section: although the authors correctly included important papers in this setting, we believe a couple of studies should be cited within the introduction (doi: 10.1111/den.13935; doi: 10.21873/invivo.11964.), only for a matter of consistency. We think it might be useful to introduce the topic of this interesting study."
We ask the authors to answer this points.
Author Response
First, we would like to thank Reviewer 3 for the valuable comments regarding our manuscript. Our responses to each of the comments are as follows.
Introduction section: although the authors correctly included important papers in this setting, we believe a couple of studies should be cited within the introduction (doi: 10.1111/den.13935; doi: 10.21873/invivo.11964.), only for a matter of consistency. We think it might be useful to introduce the topic of this interesting study.
Response:
We thank you for this suggestion. We have included these studies to the Introduction section of the revised manuscript, as follows. (page 4, lines 10)
“POCS may also be applied to percutaneous procedures [3][4].”